# Guarana, Selenium, and L-Carnitine Supplementation Improves the Oxidative Profile but Fails to Reduce Tissue Damage in Rats with Osteoarthritis

**DOI:** 10.3390/antiox14070881

**Published:** 2025-07-18

**Authors:** Aline Zuanazzi Pasinato, José Eduardo Vargas, Julia Spanhol da Silva, Joana Grandó Moretto, Cibele Ferreira Teixeira, Verônica Farina Azzolin, Ivana Beatrice Mânica da Cruz, Camile da Rosa Trevisan, Emanuele Cristina Zub, Renato Puga, Verónica Inés Vargas, Grethel León-Mejía, Rômulo Pillon Barcelos

**Affiliations:** 1Institute of Biological Sciences, University of Passo Fundo (UPF), Passo Fundo 99042-800, Rio Grande do Sul, Brazil; 167298@upf.br (A.Z.P.); romulo1604@gmail.com (R.P.B.); 2Laboratory of Inflammatory and Neoplastic Cells, Department of Cell Biology, Federal University of Paraná (UFPR), Curitiba 80060-000, Paraná, Brazil; camiletrevisan@ufpr.br (C.d.R.T.); emanuele.zub@ufpr.br (E.C.Z.); 3Graduate Program in Toxicological Biochemistry, Center for Natural and Exact Sciences (CCNE), Federal University of Santa Maria (UFSM), Santa Maria 97105-900, Rio Grande do Sul, Brazil; juliaspanholdasilva@hotmail.com; 4Graduate Program in Bioexperimentation, University of Passo Fundo (UPF), Passo Fundo 99042-800, Rio Grande do Sul, Brazil; joana_grando@yahoo.com.br (J.G.M.); ibmcruz@hotmail.com (I.B.M.d.C.); 5Graduate Program in Pharmacology, Federal University of Santa Maria (UFSM), Santa Maria 97105-900, Rio Grande do Sul, Brazil; cibelefteixeira@hotmail.com; 6Foundation of the State University of Amazonas (FUNATI), State University of Amazonas, Manaus 69050-010, Amazonas, Brazil; azzolinveronica@hotmail.com; 7Israelita Albert Einstein Hospital, São Paulo 05652-900, São Paulo, Brazil; renatopuga@gmail.com; 8Progress and Health Foundation, 41092 Sevilla, Andalucia, Spain; verovar79@gmail.com; 9Facultad de Ciencias Básicas, Centro de Investigaciones en Ciencias de la Vida (CICV), Universidad Simón Bolívar, Barranquilla 080002, Colombia; grethel.leon@unisimon.edu.co

**Keywords:** osteoarthritis, inflammation, oxidative stress, nutritional supplement, natural compounds

## Abstract

Osteoarthritis (OA) is a progressive joint disease that is commonly managed with palliative drugs, many of which are associated with undesirable side effects. This study investigated the therapeutic potential of a novel supplementation with guarana, selenium, and L-carnitine (GSC) in a rat model of chemically induced OA. Forty male Wistar rats (8–9 weeks old) received intra-articular sodium monoiodoacetate (Mia) to induce OA, and were subsequently treated with GSC. Inflammatory and oxidative stress parameters were analyzed at the end of the experiment. GSC supplementation enhanced endogenous antioxidant defenses, suggesting systemic antioxidant activity. However, no histological improvement was observed. In silico analyses indicated that Mia-induced OA may involve a complex molecular environment that GSC, at the tested dose, failed to modulate at the site of injury. Despite the limited local effects, these findings support the systemic benefits of GSC and highlight the potential of natural compound-based strategies in OA management. Given the adverse effects of conventional pharmacotherapy, the development of alternative, naturally derived treatments remains a promising avenue for future research.

## 1. Introduction

Osteoarthritis (OA) is one of the most common musculoskeletal diseases in the world and one of the main causes of disability among the elderly [1], and it may involve the hip, knee, and hand [2]. Risk factors such as age, gender, hormonal profile, bone density, obesity, and biomechanical joint changes, combined with molecular damage and an inability to effectively manage physical forces, lead to the development of this disease [3]. At the molecular level, OA is associated with higher levels of reactive oxygen species (ROS) in patients [4,5,6], inducing DNA damage mediated by interleukin-1 (IL-1) [7]. Lipid peroxidation products, such as oxidized low-density lipoprotein and nitrated compounds, are present in OA cartilage and biological fluids, linking ROS to cartilage degradation [8]. Conversely, antioxidant enzymes are reduced in OA, reinforcing oxidative stress as a key factor in pathogenesis [7,9,10]

Several natural compounds have shown promise as putative therapeutics for OA, a degenerative joint disease characterized by cartilage breakdown and inflammation. Among them, curcumin, derived from *Curcuma longa*, has demonstrated anti-inflammatory and antioxidant properties by modulating NF-κB and MAPK signaling pathways [11]. Resveratrol, a polyphenol found in grapes and red wine, exerts chondroprotective effects through the activation of SIRT1 and AMPK, thereby reducing oxidative stress and apoptosis in chondrocytes [12]. Similarly, epigallocatechin-3-gallate, the main catechin in green tea, suppresses IL-1β-induced catabolic activity and matrix degradation by inhibiting NF-κB activation [13,14]. Boswellic acids, obtained from *Boswellia serrata*, act via 5-lipoxygenase inhibition, and have been associated with pain relief and improved joint function in clinical trials [15]. In addition, ginger-derived compounds such as gingerols and shogaols modulate cyclooxygenase and lipoxygenase pathways, contributing to their anti-inflammatory efficacy [16]. Finally, avocado/soybean unsaponifiables have been reported to inhibit pro-inflammatory cytokine production and matrix-degrading enzymes, with evidence of symptom improvement and potential structure-modifying effects [17]. These compounds represent a complementary approach to conventional OA management, warranting further investigation in well-designed clinical studies [18].

In this context, the GSC formulation emerges as a promising nutritional supplement candidate for OA, due to its unique composition and synergistic effects. GSC is composed of guarana seeds (*Paullinia cupana*), which possess well-documented antioxidant, anti-inflammatory, and immunomodulatory properties [19,20,21]; Brazil nuts (*Bertholletia excelsa*), which are a natural source of selenium, an essential micronutrient involved in oxidative stress regulation [22] and immune function [23]; and red meal, which contains L-carnitine and exhibits antioxidant activity [24]. Previous studies have demonstrated that the combination of these three components in the GSC formulation exerts a more potent effect in terms of reducing oxidative stress and inflammation compared to each compound alone [25]. Moreover, safety assessments analyzed by our group confirm that this formulation is safe for human consumption [26]. In addition, a previous pilot study also conducted by our group with 28 patients showed positive results for multiple sclerosis [25]. Taken together, these findings support the inclusion of GSC among the growing list of natural products with potential benefits in the management of OA. Since, to date, no study has demonstrated the benefits of the joint use of the aforementioned components, this work aimed to identify the possible effects of GSC supplementation in the treatment of induced OA in rats in a multisystemic context and at the local site of injury.

## 2. Materials and Methods

### 2.1. Animal Model

Forty male Wistar rats (8 to 9 weeks of age) were received and housed in the Central Vivarium of the University of Passo Fundo (UPF), Brazil. The animals were kept under a controlled temperature (22 ± 2 °C) and a light/dark cycle, and had free access to water and food. The experimental protocol was approved by the Ethics Committee on the Use of Animals of the University of Passo Fundo (CEUA-UPF) under registration n˚ 26/2018.

### 2.2. GSC Supplement

The nutritional supplement used in this study consisted of 100 mg of guarana dry extract (containing 7.79% caffeine and 1.02% tannins), 50 μg of selenium in chelated form (selenium bis-glycinate 0.5%), and 400 mg of L-carnitine (as L-carnitine L-tartrate), totaling 500 mg per capsule. All compounds were acquired in 2023 from certified suppliers in São Paulo, Brazil: guarana extract from SM Pharmaceutical Enterprises Ltd., selenium from Fagron Brazil, and L-carnitine from Via Pharma. According to each manufacturer’s technical specifications and certificates of analysis, the compounds were subjected to rigorous quality control procedures, confirming the absence of heavy metals (lead, copper, antimony, cadmium, arsenic, and mercury) and microbial contamination.

The supplement was formulated and encapsulated in gelatin-based capsules by a licensed compounding pharmacy, under Good Manufacturing Practices, at the request of the research team. The capsules were then transferred to the laboratory and opened under aseptic conditions. For in vivo administration, the capsule contents were diluted in distilled water and administered via oral gavage at a dose of 44.27 mg/kg of body weight, with the animals averaging 280 g prior to treatment.

### 2.3. Experimental Design and Induction of OA with Mia

A chemical model that mimics OA in rodents can be obtained through a single intra-articular injection of Mia, a glycolysis inhibitor, which can induce progressive cartilage degeneration caused by inhibition of chondrocyte metabolism [27]. Studies demonstrate that a single injection of Mia into the knee joint is capable of inducing peripheral changes, including acute local inflammation, followed by cartilage erosion and joint disorganization, as well as increased expression of pro-inflammatory cytokines [28,29,30]. In this sense, OA induction was performed in the right posterior knee with a single intra-articular injection of Mia (2 mg/50 μL, in 0.9% NaCl) [31]. The injection was made through the suprapatellar ligament into the synovial cavity of the animals, under inhalation anesthesia with isoflurane. Both control and GSC-treated animals received injections via the same route: control animals received 50 μL of 0.9% NaCl, while treated animals received GSC. For the experiment, the rats were randomly divided into 4 groups (n = 10), designated as follows: control (control); treatment with GSC (GSC); injury with Mia (Mia); and both injury with Mia and treatment with GSC (Mia+GSC). Seven days after Mia or NaCl injection, gavage treatment was started with the GSC nutritional supplement for 30 days (1 dose/day, 5 doses/week). After 48 h had passed since the last administration, the animals were anesthetized by inhalation with isoflurane and had their trunks decapitated [32]. The liver, kidney, and brain were excised for biochemical analysis of oxidative and inflammatory parameters.

### 2.4. Biochemical Analysis of Oxidative Parameters

#### 2.4.1. Production of Thiobarbituric Acid Reactive Species (TBARS)

The levels of TBARS, a biomarker of lipid peroxidation, were measured by reacting the samples with thiobarbituric acid and quantifying malondialdehyde concentration, as described by Jentzsch et al. (1996) [33]. The reaction mixture consisted of 200 μL of homogenate, 500 μL of acetic acid buffer, 500 μL of 0.8% thiobarbituric acid (TBA), 200 μL of 8.1% sodium dodecyl sulfate (SDS), and 100 μL of H_2_O. The mixture was incubated at 95 °C for 2 h, and then read in a spectrophotometer at 532 nm. The results were expressed in nmol MDA/mg of protein.

#### 2.4.2. Production of ROS

The production of ROS by cells was determined by measuring the levels of 2′-7′- dichlorofluorescein (DCF) [34]. For the test, 50 μL of homogenate was added to a medium comprising Tris-HCl (10 mM) and 2′-7′-dichlorofluorescein diacetate (DCFH-DA) (1 mM), and incubated for 1 h in the dark. Fluorescence was evaluated (excitation at 488 nm, emission at 525 nm and aperture at 1.5 nm) and the results corrected for protein levels (adapted from Saccol et al., 2020 [35]). The results were expressed in μmol DCF/mg of protein.

#### 2.4.3. Catalase (CAT) Activity

CAT activity was determined according to the method of Nelson and Kiesow (1972) [36]. This method consists of adding 20 μL of homogenate to a mixture of 2.000 μL of 50 mM, pH 7 potassium phosphate buffer (TFK) and 100 μL of 0.3 M hydrogen peroxide (H_2_O_2_). Absorbance was recorded for 1 min (every 15 s) at 240 nm (adapted from Saccol et al., 2020 [35]). The results were expressed in μmol/mg of protein/min.

#### 2.4.4. Acetylcholinesterase (AChE)

AChE activity was estimated using the Ellman method, with acetylthiocholine iodide (ATC) as the substrate. The protocol involved adding 880 μL of 0.1 M TFK buffer to 25 μL of tissue sample, followed by incubation in a water bath at 30 °C for 2 min. Then, 50 μL of ATC 9 mM and 50 μL of DTNB 6 mM were added in order to take readings at 412 nm at the times points of 0, 60, and 120 s [37,38]. The results were expressed in mmol ATC/min/mg of protein.

#### 2.4.5. Determination of Total Thiols (T-SH)

To determine the levels of total thiols (T-SH), the method of Ellman (1959) [39] was used, with some modifications. A 200 μL volume of liver or kidney homogenate, 750 μL of 1 M, pH 7.4 potassium phosphate buffer (TFK), and 50 μL of 10 mM 5.5′-dithio-bis (2-nitrobenzoic acid) (DTNB) were added. The reaction product was measured at 412 nm. The results were expressed in μmol T-SH/mg of protein.

#### 2.4.6. Total Antioxidant Capacity (TAC)

The TAC was determined by the phosphomolybdenum method, based on the spectrophotometric determination of the reduction of Mo + 4 to Mo + 5, with subsequent formation of Mo + 5 phosphate, which shows maximum absorption at 695 nm [40]. Homogenized tissue samples (10 mg/mL), dissolved in distilled water or 1% DMSO, were combined, in an Eppendorf, with 1 mL of reagent solution (sulfuric acid 600 mM, sodium phosphate 28 mM and ammonium molybdate 4 mM). The closed tubes were incubated at 95 °C for 90 min. After cooling to room temperature, the absorbance at 695 nm was determined. The results were expressed in mmol/g of protein.

### 2.5. Histopathological Evaluation

The right posterior knees of the rats were removed, identified, fixed in 10% formaldehyde, and decalcified in 10% nitric acid. Histological sections were processed and stained for histopathology with Masson’s Trichrome, then analyzed and photographed under a microscope Zeiss Axio Lab.A1 (Carl Zeiss Microscopy GmbH, Jena, Germany) using a digital image-capture camera (AxioCam ERc5s). The Osteoarthritis Research Society International (OARSI) score was used to assess the severity of OA, analyzing the degree of cartilage histopathology (Adaptado Pritzker et al., 2006 [41]). The slides were analyzed randomly and blindly, and the degrees of histopathology are presented in Table 1.

### 2.6. Statistical Analysis

Statistical analysis was performed using the GraphPad Prism 6^®^ software, with bidirectional analysis of variance (two-way ANOVA), followed by the Sidak post-test. Data were expressed as the mean ± the standard error of the mean (S.E.M.), with statistical significance established at a *p*-value < 0.05.

### 2.7. In Silico Analysis

Gene Ontology (GO) terms related to “oxidative stress” and “response to oxidative stress” were retrieved from the Rat Genome Database, https://rgd.mcw.edu/GO/ (accessed on 16 June 2024). The associated genes were integrated with bioactive compounds from guarana seeds (*Paullinia cupana*)—theobromine, caffeine, catechin, epicatechin, theophylline, and proanthocyanidins [42,43]—along with carnitine and selenium, to construct a chemical–protein–protein interaction (CP-PPI) network using STITCH 5.0 [44], with a confidence score ≥0.7.

The resulting network was analyzed for centrality using the CentiScaPe 2.2 plugin in Cytoscape [45]. Degree centrality, indicating the number of direct connections per node, and betweenness centrality, representing the frequency with which a node lies on the shortest path between other nodes, were both calculated. Mean values were used to identify key nodes: those with scores above the network average were defined as hubs (high degree) and bottlenecks (high betweenness). Nodes with values above the mean for both parameters were further categorized as H-B nodes.

Differentially expressed genes (DEGs) were identified using data from the Gene Expression Omnibus (GEO) dataset GSE103416. This dataset comprises knee cartilage samples collected from male Wistar rats subjected to Mia-induced OA at 0, 2, 14, and 28 days after treatment. Mia (3 mg) was administered intra-articularly, and treated animals were compared to intact controls. A two-sample t-test was applied to each gene to assess differential gene expression (DEG) between groups. To evaluate statistical significance while controlling for multiple testing, a permutation-based approach with 1000 random permutations of sample labels was employed to generate empirical *p*-values. These *p*-values were subsequently adjusted using the Bonferroni correction method. In parallel, fold-change values were calculated for each gene to quantify the magnitude and direction of expression changes. Genes with Bonferroni-adjusted *p*-values below 0.01 were considered significantly differentially expressed. In addition, the Venn diagram tool was used to identify common genes between DEGs induced by OA at 0, 2, 14, and 28 days after treatment and H-B genes from the CP-PPI network. This bioinformatics tool is available at: http://bioinformatics.psb.ugent.be/webtools/Venn/ (accessed on 3 September 2024).

## 3. Results

### 3.1. Production of TBARS

Lipid peroxidation levels were significantly increased in hepatic (Figure 1A, *p* < 0.0001), renal (Figure 1B, *p* = 0.0158), and cerebral (Figure 1C, *p* = 0.0204) tissues in the Mia group compared to the control. However, treatment with GSC (Mia+GSC group) restored TBARS levels to values comparable to those of the control group in all three tissues (Figure 1A–C).

### 3.2. ROS Production

ROS production in the liver (Figure 2A) was increased in both the GSC (*p* = 0.0286) and Mia+GSC (*p* < 0.0001) groups compared to the control group. A similar effect was observed in the kidneys (Figure 2B); however, the Mia group also showed an increase in ROS production compared to the control (*p* = 0.0184). In the brain (Figure 2C), ROS levels were elevated in the Mia group compared to the control (*p* = 0.0040), but GSC treatment prevented this increase.

### 3.3. CAT Activity

CAT activity in the liver (Figure 3A) was significantly reduced in the GSC (*p* = 0.0019) and Mia+GSC (*p* = 0.0002) groups compared to the control group. In the kidneys (Figure 3B), a decrease in enzyme activity was observed in the Mia+GSC group relative to the control (*p* = 0.0345). In the brain (Figure 3C), no significant difference was found between the control and Mia+GSC groups.

### 3.4. AChE Quantification

AChE enzyme activity in the liver (Figure 4A) was significantly increased in the Mia group compared to in the control, GSC, and Mia+GSC groups (*p* = 0.0015, *p* = 0.0003, *p* = 0.0183, respectively). GSC treatment was able to reverse this increase, restoring AChE activity to levels similar to those of the control group. No significant changes in enzyme activity were observed in renal tissue (Figure 4B). In the brain (Figure 4C), the Mia+GSC group exhibited a reduction in AChE activity compared to the Mia group (*p* = 0.0058), reaching levels comparable to those of the control group.

### 3.5. Determination of Total T-SH

Total thiol levels were not significantly different in hepatic (Figure 5A) and renal (Figure 5B) tissues. However, in the brain (Figure 5C), the GSC group showed an increase in total thiol levels compared to the control group (*p* = 0.0244), while the levels of the Mia+GSC group remained similar to those of the control.

### 3.6. Measure of TAC

Analysis of liver TAC (Figure 6A) revealed a significant increase in the GSC group compared to in the control group (*p* = 0.009). Additionally, GSC treatment normalized TAC levels in the Mia+GSC group, making them comparable to those of the control. In the kidneys (Figure 6B), CT values were reduced in the Mia group relative to the control (*p* = 0.0317), while the Mia+GSC group showed values similar to those of the control. In the brain (Figure 6C), CT was increased in the Mia+GSC group compared to the Mia group (*p* = 0.0339), with levels also matching those of the control group.

### 3.7. Systems Biology Analysis and Histological Evaluation

To investigate potential interactions between the GSC mix formulation and pathways associated with oxidative stress, a chemical–protein–protein interaction (CP-PPI) network consisting of 274 nodes and 1404 edges was predicted (Figure 7A). Centrality analysis was performed, identifying 49 H-B nodes (Figure 7B). Interestingly, Venn diagram analysis revealed an uneven distribution of differentially expressed genes (DEGs) among these H-B nodes during Mia-induced osteoarthritis at 0, 2, 14, and 28 days post treatment (Figure 7C). Only three genes—*Prdx1*, *Nfkb1*, and *Bmp4*—were found to be differentially expressed under the analyzed conditions. However, the expression levels of these genes varied depending on the timing following Mia administration (Figure 7D). Histopathological analysis (Figure 7E) demonstrated cartilage denudation and subchondral bone degeneration in Mia-treated animals, which corresponded to significantly higher OARSI scores in these groups (Figure 7F), with no observable alterations following treatment with the GSC mix.

## 4. Discussion

The effects of caffeine [46], selenium [47], and L-carnitine [48] have been extensively characterized in previous studies. In the present study, we aimed to investigate the antioxidant mechanisms and potential therapeutic effects of the complete guarana extract combined with selenium and L-carnitine in the context of OA, rather than examining the effects of these constituents individually. This approach was reflected in the actual experimental conditions, as the animals were administered the whole extract, whose absorption, pharmacokinetics, and systemic distribution differ substantially from those of isolated compounds. Our results confirmed the antioxidant effects of the supplementation, particularly illustrated by a reduction in TBARS levels in OA animals given the GSC supplementation. This effect can be attributed to caffeine’s capacity to inhibit lipid peroxidation and decrease ROS production [49]. In contrast, the non-supplemented OA groups exhibited elevated TBARS levels, likely as a result of inflammation-induced ROS production, which contributes significantly to cartilage and bone degradation [50], as well as neutrophil degranulation and the release of harmful enzymes [51]. The established link between inflammation and oxidative stress is particularly critical, considering that the brain, with its high oxygen consumption and limited antioxidant defenses, is especially vulnerable to ROS-induced damage [52]. Notably, our results indicated a reduction in brain ROS levels in OA animals supplemented with GSC, supporting the neuroprotective effects associated with caffeine and L-carnitine [53,54]. However, the observed increase in ROS levels in the liver and kidney tissues of supplemented animals might reflect the complex dose-dependent effects of caffeine, which can manifest both beneficial and adverse outcomes [49].

To modulate oxidative damage, the body relies on antioxidant defense mechanisms, among which enzymatic systems like CAT play a vital role. Our findings revealed tissue-specific variations in CAT activity: GSC supplementation resulted in increased CAT activity in brain tissue, indicating a protective role, but it was associated with reduced CAT levels in other tissues. Such variability underscores the complex regulatory mechanisms controlling antioxidant enzyme activity across different tissues [55]. In the context of inflammatory states, acetylcholine—a neurotransmitter essential for neuromuscular communication and immune modulation—undergoes rapid degradation mediated by AChE. Elevated AChE activity has been linked to inflammatory conditions [56]. In our study, OA animals exhibited increased AChE levels; conversely, GSC supplementation led to a reduction in AChE activity, reinforcing the protective effects of selenium on immune function and cellular homeostasis [57]. Moreover, thiols serve as effective biomarkers of oxidative stress, providing insights into the balance between free radical production and antioxidant defenses [58]. While our results showed no significant alterations in total TS-H levels among injured groups, the increase in TS-H levels in GSC-supplemented animals supports the beneficial effects of its components in enhancing immune function and mitigating oxidative stress [59]. Total antioxidant capacity (TAC) is another key indicator of systemic antioxidant defense efficiency [60]. Previous studies have established a positive correlation between selenium supplementation and increased TAC, alongside decreased malondialdehyde levels [58]. Consistent with these findings, our study reported elevated TAC in GSC-supplemented groups, further confirming the antioxidant properties attributed to caffeine [49,61] and underscoring the potential protective effect of the combined use of guarana, selenium, and L-carnitine in the context of OA. It is important to note that many natural compounds, such as harpagoside [62] and avocado/soybean unsaponifiables [63], primarily exhibit local anti-inflammatory or chondroprotective effects, with limited systemic antioxidant activity.

Despite this systemic benefit, the findings revealed that GSC supplementation failed to ameliorate local tissue damage resulting from OA. This implies that reducing oxidative stress alone is inadequate to counteract the structural deterioration associated with the disease. The in silico analyses indicated a complex molecular environment in Mia-induced OA that was not sufficiently modulated by GSC under the tested dose and timeframe. Although there was documented improvement in oxidative parameters in supplemented animals, histopathological analysis indicated no significant improvement in cartilage preservation or OARSI scores in GSC-treated animals. Our results also highlighted that only three central genes—Prdx1, Nfkb1, and Bmp4—were differentially expressed during Mia-induced osteoarthritis, demonstrating a limited overlap in DEGs related to Mia-induced OA. This suggests dose-dependent variability in pathway-level responses to Mia-induced OA. However, the centrality of Prdx1 and Nfkb1 underscores their relevance in OA-related oxidative stress, positioning them as potential biomarkers or therapeutic targets.

These findings can also be used to guide future optimization of the GSC formulation, dosing strategies, and timing of interventions, emphasizing that the lack of measurable protective effects does not equate to ineffectiveness, but rather reflects the complexity of OA pathophysiology and the necessity for refined therapeutic approaches that target early oxidative responses. One possible explanation for the lack of histological improvement is that, while oxidative stress is known to contribute to OA pathogenesis, other factors—such as persistent biomechanical stress, chronic inflammation, and catabolic enzyme activity—may play more significant roles in cartilage degradation. GSC’s inability to counter these mechanisms highlights the multifaceted nature of OA. Furthermore, the intricate interplay between oxidative stress and inflammatory pathways suggests that although antioxidants like those in GSC may reduce oxidative damage, they may not directly inhibit the inflammatory mediators responsible for tissue destruction. Given the adverse effects of conventional pharmacotherapy, the development of alternative, naturally derived treatments remains a promising avenue for future research in OA management.

## 5. Conclusions

In summary, our results indicate that supplementation with guarana, selenium, and L-carnitine effectively improves oxidative parameters in OA; however, it does not prevent joint tissue damage. This highlights the complexity of OA pathogenesis and suggests that targeting oxidative stress alone is insufficient to achieve structural protection of the joints.

It is important to acknowledge certain limitations of our study. Only a single dose of each compound was tested, and the treatment duration was relatively short, which may have limited the possibility of detecting structural benefits in cartilage within this progressive disease model. Moreover, although the study focused primarily on oxidative stress, future investigations should also explore inflammatory pathways and cartilage matrix degradation. Such research is essential to fully elucidate the mechanisms underlying OA and to evaluate combined therapeutic approaches that address oxidative stress, inflammation, and biomechanical factors contributing to disease progression.

## Figures and Tables

**Figure 1 antioxidants-14-00881-f001:**
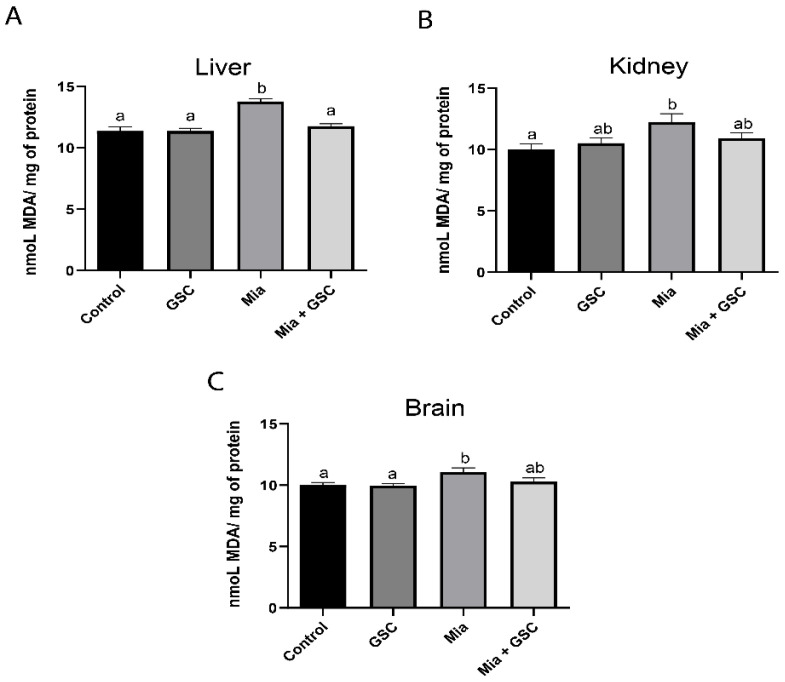
The effect of GSC supplementation on the production of TBARS in the liver (**A**), kidney (**B**), and brain (**C**) of animals with osteoarthritis. The animals were divided into four groups (n = 10): control, GSC, Mia, and Mia+GSC. The data are expressed as the mean ± S.E.M. Means with different letters indicate significant differences (*p* < 0.05).

**Figure 2 antioxidants-14-00881-f002:**
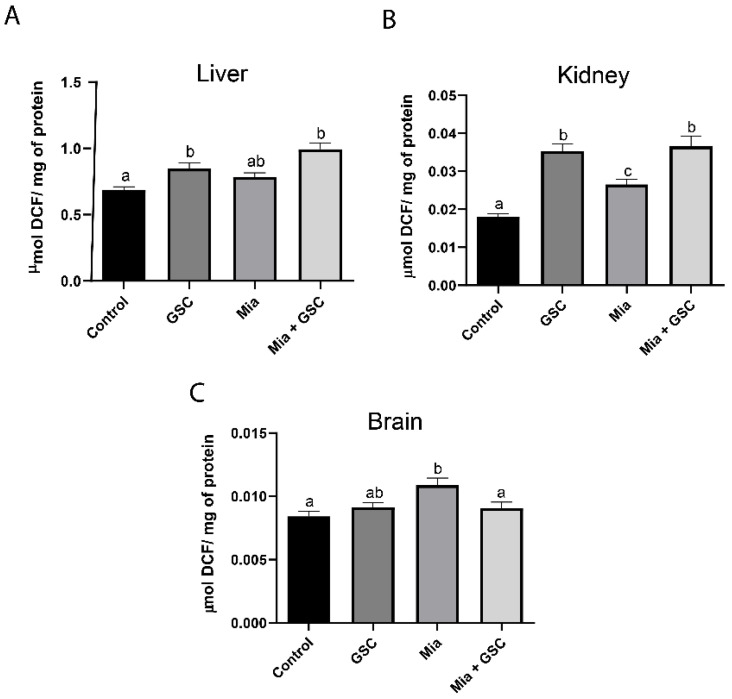
The effect of GSC supplementation on ROS production in the liver (**A**), kidney (**B**), and brain (**C**) of animals with osteoarthritis. The animals were divided into four groups (n = 10): control, GSC, Mia, and Mia+GSC. The data are expressed as the mean ± S.E.M. Means with different letters indicate significant differences (*p* < 0.05).

**Figure 3 antioxidants-14-00881-f003:**
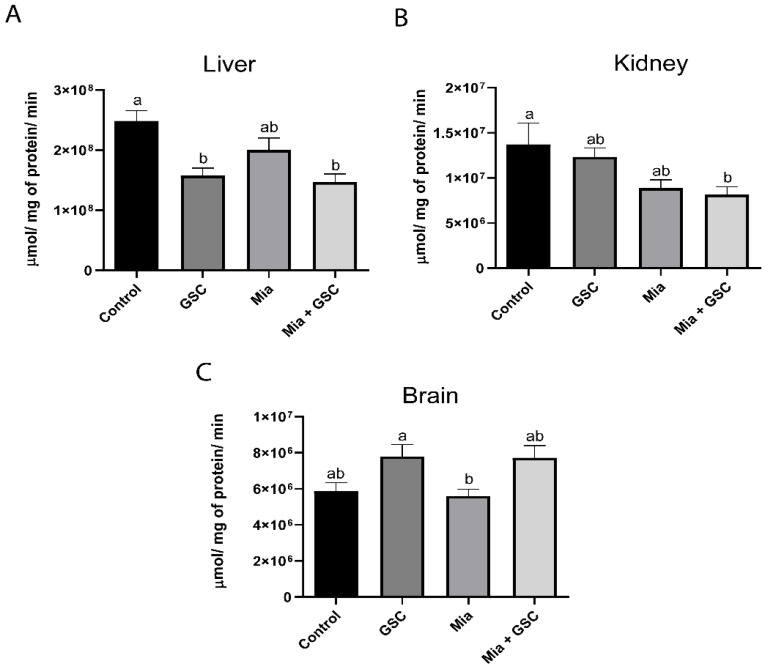
The effect of GSC supplementation on CAT activity in the liver (**A**), kidney (**B**), and brain (**C**) of animals with osteoarthritis. The animals were divided into four groups (n = 10): control, GSC, Mia, and Mia+GSC. The data are expressed as the mean ± S.E.M. Means with different letters indicate significant differences (*p* < 0.05).

**Figure 4 antioxidants-14-00881-f004:**
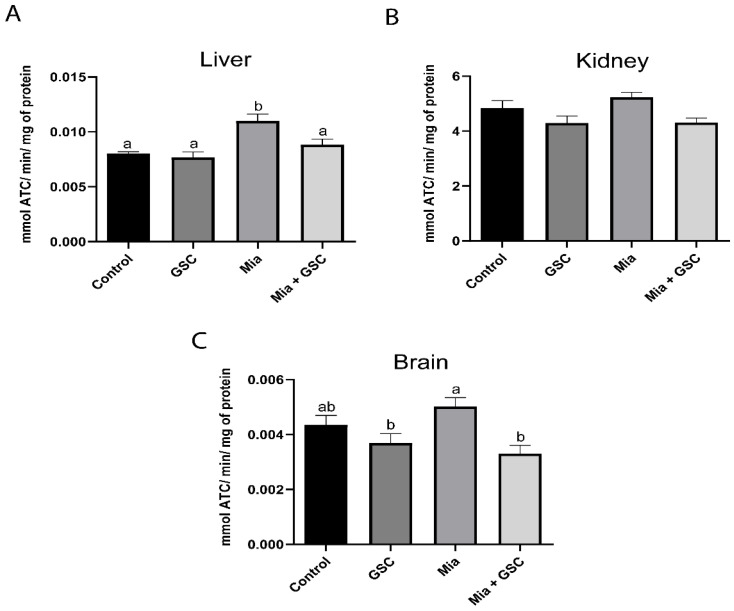
The effect of GSC supplementation on the activity of the enzyme AChE in the liver (**A**), kidney (**B**), and brain (**C**) of animals with osteoarthritis. The animals were divided into four groups (n = 10): control, GSC, Mia, and Mia+GSC. The data are expressed as the mean ± S.E.M. Means with different letters indicate significant differences (*p* < 0.05).

**Figure 5 antioxidants-14-00881-f005:**
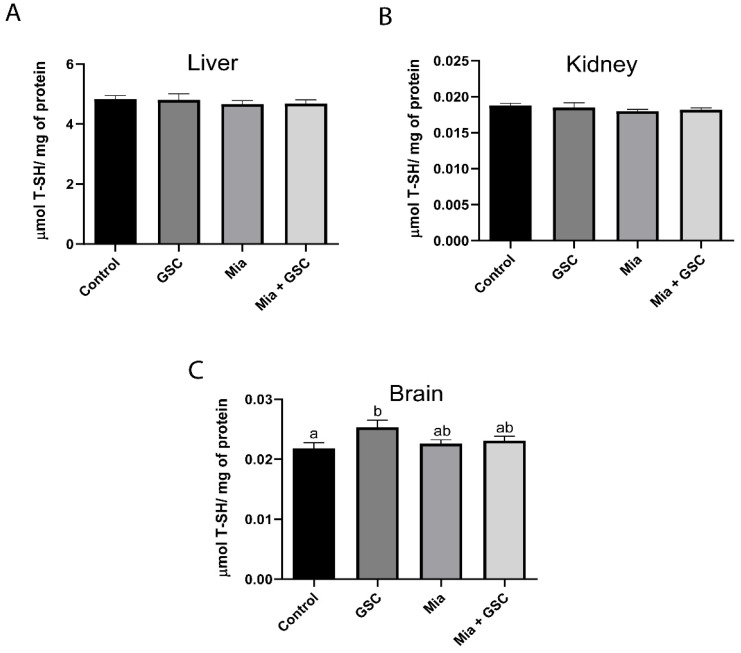
The effect of GSC supplementation on total T-SH levels in the liver (**A**), kidney (**B**), and brain (**C**) of animals with osteoarthritis. The animals were divided into four groups (n = 10): control, GSC, Mia, and Mia+GSC. The data are expressed as the mean ± S.E.M. Means with different letters indicate significant differences (*p* < 0.05).

**Figure 6 antioxidants-14-00881-f006:**
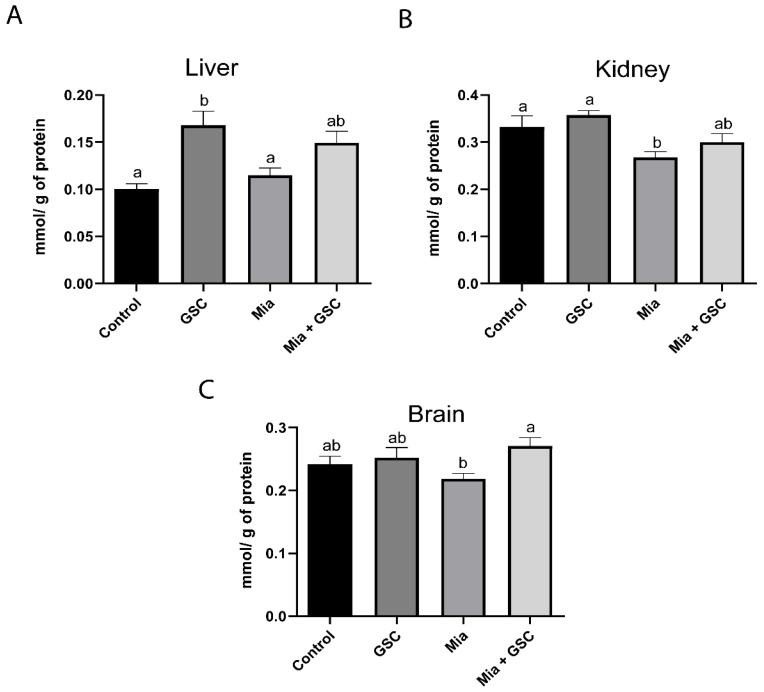
The effect of GSC supplementation on TAC in the liver (**A**), kidney (**B**), and brain (**C**) of animals with osteoarthritis. The animals were divided into four groups (n = 10): control, GSC, Mia, and Mia+GSC. The data are expressed as the mean ± S.E.M. Means with different letters indicate significant differences (*p* < 0.05).

**Figure 7 antioxidants-14-00881-f007:**
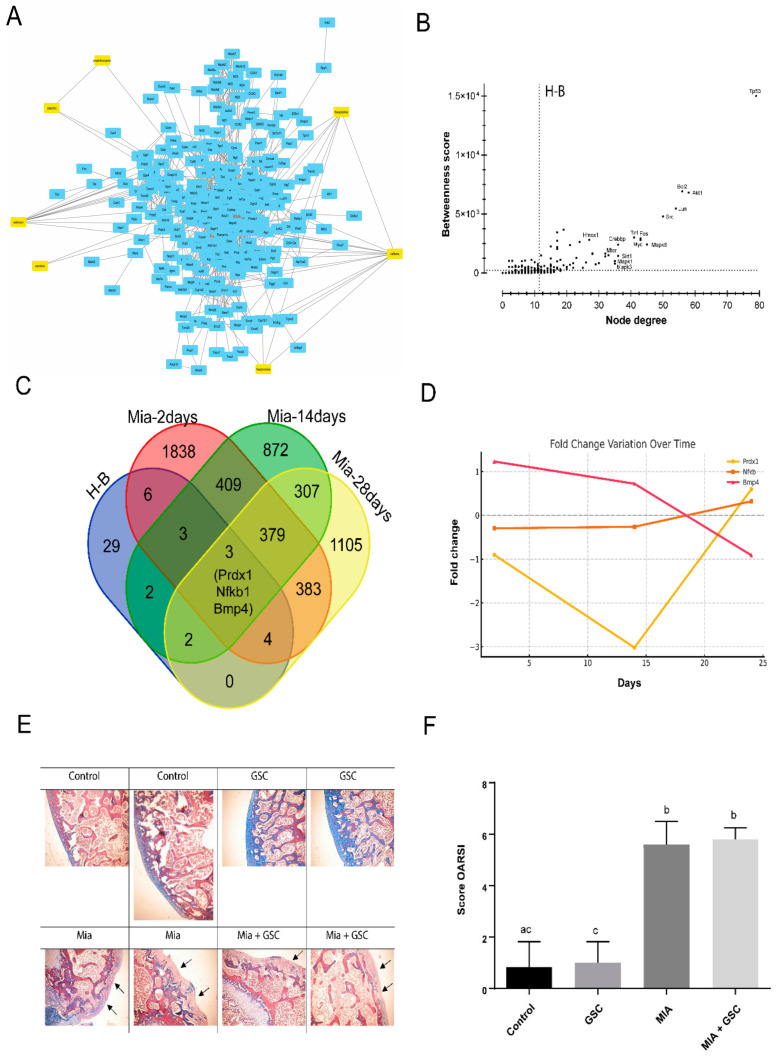
The interaction network, gene expression analysis related to oxidative stress, and histological evaluation in Mia-induced osteoarthritis and GSC treatment. (**A**) The CP-PPI network constructed using STITCH 5.0, integrating bioactive compounds from the GSC mix with oxidative stress-related genes. Compounds are highlighted in yellow. (**B**) Centrality analysis performed using the CentiScaPe 2.2 plugin in Cytoscape. The H-B nodes exhibit both degree and betweenness centrality scores above the network average. (**C**) A Venn diagram showing the distribution of differentially expressed genes (DEGs) among H-B nodes at 0, 2, 14, and 28 days after Mia administration. (**D**) Fold-change values of the three DEGs (Prdx1, Nfkb1, and Bmp4) at the indicated time points following Mia treatment. (**E**) Representative histological sections of knee cartilage showing cartilage denudation and subchondral bone degeneration in Mia-treated animals, with or without GSC treatment. Animals were divided into four groups (n = 10): control, GSC, Mia, and Mia+GSC. Blue staining indicates the presence of collagen, while black arrows denote areas of collagen loss. The data are expressed as the mean ± S.E.M. Groups with different letters are significantly different (*p* < 0.05). (**F**) Histopathological scoring using the OARSI grading system.

**Table 1 antioxidants-14-00881-t001:** Histological grading of cartilage degeneration.

Degree	Characteristic
0	Surface and cartilage with intact morphology
1	Intact surface
2	Discontinuity surface
3	Vertical cracks
4	Erosion
5	Denudation
6	Deformation

## Data Availability

The data presented in this study are available upon request from the corresponding author.

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
