# Peer review of "Guarana, Selenium, and L-Carnitine Supplementation Improves the Oxidative Profile but Fails to Reduce Tissue Damage in Rats with Osteoarthritis"

_antioxidants, 2025, doi:10.3390/antiox14070881_

Round 1
Reviewer 1 Report
The manuscript: Guarana, Selenium, and L-Carnitine Supplementation Improves the Oxidative Profile but Fails to Reduce Tissue Damage in Rats with Osteoarthritis investigates the effects of a guarana, selenium, and L-carnitine supplement osteoarthritis. The study is relevant but there are errors that should be improved before publication
There are numerous language errors:
- “...the development of alternative, naturally derived treatments remain a promising avenue…” better “remains”,
- “...combination of these three components...exerts a more potent effect…” Consider rephrasing,
- “AO” or should be “OA”,
- “Histhological evaluation” should be Histological,
- “The injections was made...” was one or should be “were”,
There are other issues that can be improved:
- Add the results section quantitative data that are shown in the text
- “did not mitigate local tissue damage” there is the inconsistency with conclusions, probably should be "no evidence of histological improvement",
- “the studied tissue” usually we say “tissue sample”
- “CT” used instead of TAC in some places
- Clarify GSC tissue-specific effects
Author Response
We sincerely thank the reviewers for their thorough and constructive feedback. We have carefully addressed each comment and revised the manuscript accordingly. Below, we provide point-by-point responses to each suggestion.
Reviewer 1
- Comment: There are numerous language errors.
Response: We appreciate the reviewer’s attention to language. We have thoroughly revised the manuscript for grammar and clarity. Specific corrections include: 'remain' to 'remains'; rephrasing of the combination sentence; correction from 'AO' to 'OA'; correction of 'Histhological' to 'Histological'; and correction from 'was' to 'were', when necessary.
- Comment: Add the results section quantitative data that are shown in the text.
Response: We have revised the results section to include all relevant quantitative data referenced in the text to improve clarity and reproducibility.
- Comment: 'did not mitigate local tissue damage' is inconsistent with conclusions.
Response: This sentence has been revised to 'no histological improvement was observed' to better reflect the findings.
- Comment: 'the studied tissue' should be 'tissue sample'.
Response: Corrected as suggested.
- Comment: 'CT' used instead of TAC in some places.
Response: All instances have been revised to consistently use 'TAC' (Total Antioxidant Capacity).
- Comment: Clarify GSC tissue-specific effects.
Response: We clarified in the discussion that GSC supplementation had tissue-specific responses regarding antioxidant activity and ROS modulation.
We hope that these revisions address the reviewers’ concerns and improve the clarity, quality, and scientific rigor of our manuscript. Thank you once again for the insightful feedback.
Sincerely,
Dr. José Eduardo Vargas
on behalf of all co-authors.
Corresponding Author: jose.vargas@ufpr.br
Reviewer 2 Report
This article presents the implications of guarana, selenium, and L-Carnitine supplementation in rat model of chemically induced OA. The topic is relevant, but several major deficiencies have been identified, both in content and form. These need to be addressed based on the specific recommendations below:
- Consecutive multiple references such as [4–6], [7, 9, 10], etc., should be avoided as they make it difficult to correlate the presented information with the cited sources. Instead, individual citations should be placed directly after each specific piece of information, rather than grouped at the end of a paragraph. Please revise this throughout the entire manuscript.
- Abbreviations should be defined at their first occurrence, and then only the abbreviated form should be used. However, if an abbreviation is used only once, the full term should be written out instead (e.g., SOD, CAT, GPX, PON1, MAPK, etc.). Please revise the manuscript based on these principles, keeping in mind that the abstract is treated separately from the main text in this respect.
- The full Latin names of plant species must be written in italics.
- What was the capsule shell made of? Depending on its composition, it can influence both the release and absorption of the active substances.
- “In this sense, OA induction was performed in the right posterior knee with a single intra-articular injection of Mia (2 mg/50 μL in 0.9% NaCl) [30].” Why is a reference needed here? All relevant data should be included to ensure reproducibility. The same applies to reference [31].
- Figure titles must comply with the formatting and editorial style provided in the journal’s template.
- The discussion section should be improved by comparing the data obtained with other similar studies, even if different phytocompounds were tested for the same purpose or if the experimental design was not exactly the same. These evaluations are essential in this context.
- Updating the references is mandatory. There are many outdated references relative to the current context and methodological standards (e.g., 1996, 2006, 1999, 1999, 1961, 1972, 2009, 2011, 2004, 2009, 2012, 2016, 2007, 2008, 2010, 2015), which is too many in proportion to the total number of references.
- A 35% similarity index is far too high, especially considering that the article is classified as original research.
Author Response
We sincerely thank the reviewers for their thorough and constructive feedback. We have carefully addressed each comment and revised the manuscript accordingly. Below, we provide point-by-point responses to each suggestion.
Reviewer 2
- Comment: Avoid consecutive multiple references such as [4–6], etc.
Response: We revised the manuscript so that references are now cited individually after each specific piece of information, when necessary.
- Comment: Abbreviations should be defined at first occurrence...
Response: We have revised the manuscript accordingly, defining abbreviations at first use and removing abbreviations used only once.
- Comment: Latin names must be in italics.
Response: Corrected throughout the manuscript.
- Comment: What was the capsule shell made of?
Response: The capsule shell was gelatin-based, which is not expected to interfere significantly with compound release or absorption.
- Comment: Why is a reference needed for MIA injection?
Response: We appreciate the reviewer’s question. In our Methods section, we included references to support the use of monosodium iodoacetate (MIA) as a well-established chemical model to induce osteoarthritis (OA) in rodents. While the procedure itself is commonly used, the inclusion of references is important to justify the choice of dose, site of injection, mechanism of action (inhibition of chondrocyte metabolism), and to contextualize the expected pathological features such as cartilage degradation, joint inflammation, and cytokine production. These citations also serve to demonstrate adherence to protocols validated in the literature and enhance the reproducibility of the model. For these reasons, we believe that referencing previous studies is both appropriate and necessary.
- Comment: Figure titles must follow the journal’s style.
Response: We reformatted all figure titles and legends to comply with the journal’s editorial style.
- Comment: Improve discussion with comparisons to other studies.
Response: We expanded the discussion to include comparisons with studies of other phytocompounds used in OA models.
- Comment: Update references. Many are outdated.
Response: We updated the manuscript by replacing or supplementing older citations, except for classical methods, which remain relevant.
- Comment: 35% similarity is too high.
Response: We thank the reviewer for this important observation. To address the concern regarding similarity, we conducted an in-depth plagiarism analysis using specialized software, Plagium Premium 1.5.5, with the “deep analysis” option enabled. The detailed report indicated a similarity index of only 12.7%, with the majority of matches related to standardized terminology and the names of statistical tests, which are unavoidable in scientific writing.
It is important to highlight that the reference section was excluded from the analysis to avoid false-positive similarity detection, as bibliographic entries often inflate similarity percentages without representing actual content overlap. A copy of the full plagiarism report has been attached to this submission for verification, should the editorial team wish to review it.
We remain committed to ensuring the originality and integrity of our work and hope this clarification adequately addresses the concern.
We hope that these revisions address the reviewers’ concerns and improve the clarity, quality, and scientific rigor of our manuscript. Thank you once again for the insightful feedback.
Sincerely,
Dr. José Eduardo Vargas
on behalf of all co-authors.
Corresponding Author: jose.vargas@ufpr.br

Reviewer 3 Report
This manuscript describes the investigation of a supplementation with guarana, selenium, and L-carnitine (GSC) in a rat model of chemically induced OA. Inflammatory and oxidative stress parameters were analyzed at the end of the experiment. The authors found out that GSC supplementation enhanced endogenous antioxidant defenses but did not mitigate local tissue damage caused by OA. The experiments were designed reasonably well for the purpose, but more controls may be needed to support some of the authors’ claims.
A few concerns for the authors:
- On page 2, line 6 of the 1st paragraph, “AO is associated with higher levels de ROS in patients”. Should it be “higher levels of ROS”?
- On page 3, in “2.1. Animal Model”, the body weights of the tested animal before the experiment started will need to be indicated.
- On page 3, in the preparation of GSC supplement, more information about Guarana, like its origin, collecting time, quality control etc. should be included. Also, the origins of Selenium and L-carnitine need to be indicated too.
- On page 13, in the 1st paragraph of “4. Discussion:”, the authors attributed the “reduction in brain ROS levels in OA animals supplemented with GSC” to “caffeine and L-carnitine”. It would be more grounded if caffeine and L-carnitine were individually tested side by side the GSC supplement.
- On page 13, in the 2nd paragraph of “4. Discussion:”, the authors concluded that “the protective effects of selenium on immune function and cellular homeostasis” based on the reduced AChE activity in the GSC treated group. Maybe in the AChE experiment, a control group with only selenium treatment would be necessary.
Author Response
We sincerely thank the reviewers for their thorough and constructive feedback. We have carefully addressed each comment and revised the manuscript accordingly. Below, we provide point-by-point responses to each suggestion.
Reviewer 3
- Comment: 'AO is associated with higher levels de ROS' should be 'of ROS'.
Response: Corrected as suggested.
- Comment: Indicate body weights of animals before experiment.
Response: We added the animals’ average initial body weights to the “GSC Supplement” section.
- Comment: Provide more detail about GSC supplement source.
Response: We expanded this section to describe the origin, collection time, and quality control of each compound used in GSC.
- Comment: On page 13, in the 1st paragraph of “4. Discussion:”, the authors attributed the “reduction in brain ROS levels in OA animals supplemented with GSC” to “caffeine and L-carnitine”. It would be more grounded if caffeine and L-carnitine were individually tested side by side the GSC supplement
Response: We appreciate the reviewer’s comment. However, testing the individual components in isolation was not the objective of this study. Previous reports have already addressed the effects of caffeine and L-carnitine separately. Our goal was to investigate the antioxidant mechanisms of the full extract, rather than those of its isolated constituents. This is because the animals consumed the complete extract, and its absorption, pharmacokinetics, and systemic distribution differ significantly from those of isolated.
We have now addressed this point in the revised discussion to clarify our rationale and the scope of the experimental design.
- Comment: On page 13, in the 2nd paragraph of “4. Discussion:”, the authors concluded that “the protective effects of selenium on immune function and cellular homeostasis” based on the reduced AChE activity in the GSC treated group. Maybe in the AChE experiment, a control group with only selenium treatment would be necessary.
Response: While informative, our design aimed to assess the combined formulation’s effect. In addition, our discussion of selenium’s mechanism of action remains relevant: once the extract is metabolized and the molecule reaches its target, selenium can exert its known effects at the cellular membrane, as previously described in the literature (Int J Nanomedicine. 2015 Sep 4;10:5663-70. doi: 10.2147/IJN.S87190; Toxicol Rep. 2015;2:961-967. doi: 10.1016/j.toxrep.2015.06.01).
We hope that these revisions address the reviewers’ concerns and improve the clarity, quality, and scientific rigor of our manuscript. Thank you once again for the insightful feedback.
Sincerely,
Dr. José Eduardo Vargas
on behalf of all co-authors.
Corresponding Author: jose.vargas@ufpr.br
Round 2
Reviewer 2 Report
The authors have significantly improved the manuscript based on the suggestions received.
The authors have significantly improved the manuscript based on the suggestions received.
Reviewer 3 Report
My major concerns have been addressed in the resubmitted manuscript. No more concerns for the authors.
My major concerns have been addressed in the resubmitted manuscript. No more concerns for the authors.